

# Superconductivity in the Hubbard model: a hidden-order diagnostics from the Luther-Emery phase on ladders

**Luca F. Tocchio[1][⋆], Federico Becca[2] and Arianna Montorsi[1]**

**1** Institute for Condensed Matter Physics and Complex Systems, DISAT,
Politecnico di Torino, I-10129 Torino, Italy
**2** Dipartimento di Fisica, Università di Trieste, Strada Costiera 11, I-34151 Trieste, Italy

⋆ luca.tocchio@polito.it

## Abstract

Short-range antiferromagnetic correlations are known to open a spin gap in the repulsive Hubbard model on ladders with $M$ legs, when $M$ is even. We show that the spin gap originates from the formation of correlated pairs of electrons with opposite spin, captured by the hidden ordering of a spin-parity operator. Since both spin gap and parity vanish in the two-dimensional limit, we introduce the fractional generalization of spin parity and prove that it remains finite in the thermodynamic limit. Our results are based upon variational wave functions and Monte Carlo calculations: performing a finite size-scaling analysis with growing $M$, we show that the doping region where the parity is finite coincides with the range in which superconductivity is observed in two spatial dimensions. Our observations support the idea that superconductivity emerges out of spin gapped phases on ladders, driven by a spin-pairing mechanism, in which the ordering is conveniently captured by the finiteness of the fractional spin-parity operator.



# 1  Introduction

Since the discovery of high-temperature cuprate superconductors, it became clear that anti-ferromagnetic correlations are crucial to the onset of superconducting behavior in low dimensions [1]. The phenomenon was observed upon doping an antiferromagnetic insulator, and led to the proposal of a resonating valence bond state of singlet pairs formulated by Anderson [2]. Already in one-dimensional (1D) strongly-correlated materials, a similar superconducting state can be found. Here, the peculiar spin-charge separation allows the opening of the spin gap in the presence of gapless charge excitations, leading to the so-called Luther-Emery (LE) phase. For an appropriate behavior of the charge channel, singlet superconducting correlations may decay to zero slower than any other correlation; in this case the LE phase is often dubbed "superconducting", meaning that superconducting correlations are dominant. Such phase turns out to be characterized by the hidden ordering of a non-local operator, the spin parity operator [3–5], which describes the presence of pairs of electrons with opposite spin, that have a finite correlation length. Non-local operators can be measured via quantum gas microscopy in fermionic cold atom systems [6,7].

Because of the characteristic antiferromagnetic behavior of undoped cuprates, the Hubbard model has immediately become the reference model for their investigation. In the following, we will consider the Hubbard model on ladders with $M$ legs and $L_x$ rungs, which is defined by:

$$\mathcal{H} = -t \sum_{\langle R,R'\rangle,\sigma} c^\dagger_{R,\sigma} c_{R',\sigma} + \text{h.c.} + U \sum_R n_{R,\uparrow} n_{R,\downarrow} \,, \tag{1}$$

where $c^\dagger_{R,\sigma}$ ($c_{R,\sigma}$) creates (destroys) an electron with spin $\sigma$ on site $R$ and $n_{R,\sigma} = c^\dagger_{R,\sigma} c_{R,\sigma}$ is the electronic density per spin $\sigma$ on site $R$. In the following, we indicate the coordinates of the sites with $R = (x, y)$. The nearest-neighbor intrachain ($t_\parallel$) and interchain ($t_\perp$) hopping amplitudes are taken to be equal, $t_\parallel = t_\perp = t$; $U$ is the on-site Coulomb interaction. The electronic density is fixed and given by $n = N_e/L$, where $N_e$ is the number of electrons (with vanishing total magnetization) and $L = M \times L_x$ is the total number of sites.

In 1D, the Hubbard model is known to exhibit a LE phase only for attractive interaction, i.e., $U < 0$ [8]; by contrast, in two spatial dimensions (2D) the Hubbard model is believed to support $d$-wave superconductivity for repulsive interaction in a large range of doping values [9–16]. A possible precursor of such phase in quasi-one-dimensional lattices could be the spin gapped phase, which is known to characterize the repulsive case on ladders with an even number of legs. In this respect, recent studies [17] confirm that the insulating state at half filling, which is the counterpart of the LE phase in the charge sector, has the same microscopic structure passing from the one to the two dimensional case by increasing the number of legs $M$ of a ladder. A similar behavior has been also observed within the bosonic Hubbard model [18]. The appearance of the insulating phase is signaled by the hidden ordering of a charge-parity operator, which has to be properly normalized to $M$ in order to remain finite up to the 2D limit [18].

For generic values of $t_\parallel$ and $t_\perp$, ladder systems have been an object of intense research due to the fact that they can be assessed both by analytical tools at small values of the Coulomb repulsion, i.e., bosonization and renormalization group, and by numerical approaches, like density-matrix renormalization group (DMRG). In particular, the two-leg ladder has been thoroughly studied by DMRG [19–21]. The system is a spin gapped insulator at half filling, with the spin gap persisting also at finite doping for $t_\perp/t_\parallel < 2$. In the case of interest for this paper, i.e., $t_\perp/t_\parallel = 1$, the spin gap closes exactly at quarter filling, in agreement with the $U = 0$ one-band to two-band transition. In the spin-gapped region, superconducting correlations are present with a power-law decay. On general grounds, it is known that they are dominant when decaying with a power law with exponent smaller than 1. Previous DMRG calculations [19–21]

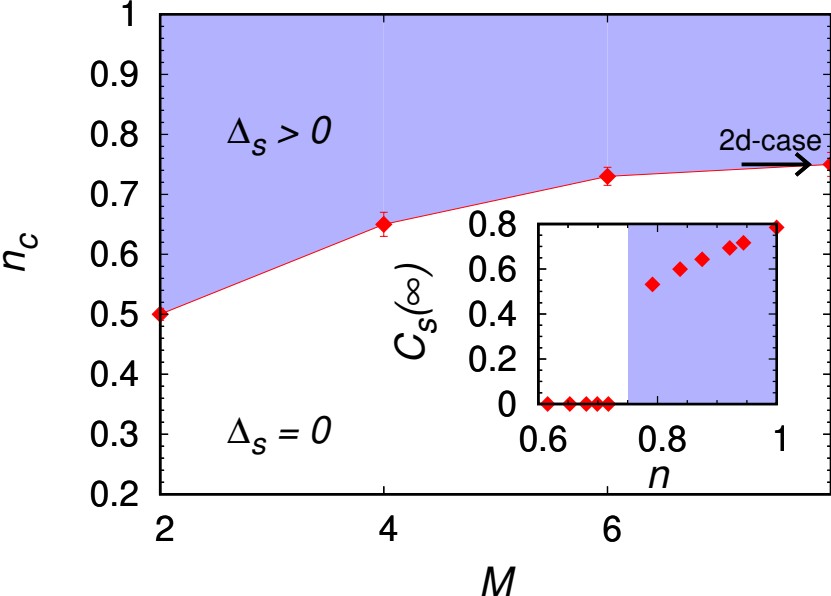

Figure 1: Main panel: Critical density $n_c$ for the opening of the spin gap in ladder systems, as a function of the number of legs $M$, compared with the density at which superconducting correlations start to develop in a fully 2D numerical calculation, see Ref. [16]. Inset: fractional spin parity $C_s(\infty)$ in the 2D limit, as a function of the density $n$. Here, the blue region has finite superconducting correlations. All data are shown at $U/t = 8$.

suggested that this was not the case, and density-density correlations are dominant; instead, more recently, it was proved by accurate DMRG investigations on large clusters that superconducting correlations dominate in the slightly doped region [22]. As for the presence of the spin gap, similar results have been obtained by the weak-coupling approach of Ref. [23], where the doped two-leg ladder falls into the LE universality class in a large region of the phase diagram. Within a weak-coupling approach, the existence of a spin gap had been already postulated in Refs. [24–26]. Finally, the presence of dominating superconducting correlations in the two-leg ladder has been also suggested by bosonization [27].

The three-leg and four-leg case have been also addressed by weak-coupling approaches [28]. The phase diagram is very rich and with a strong dependence on boundary conditions; however some common features may be observed, like the $d$-wave nature of the electron pairing and the odd-even effect in the spin gap. The latter one means that a spin gap is present at half-filling (and also at finite doping) only when the number of legs is even, as originally observed for spin models [29,30] and later derived with bosonization techniques for the Hubbard model at weak coupling [31]. Finally, the six-leg case has been investigated by DMRG and by other numerical approaches in order to assess the stability of striped phases with charge and spin modulations in the strong coupling limit [32–34].

In this paper, by extrapolating to the 2D limit the variational Monte Carlo results for the fractional spin parity on two-, four-, and six-leg ladders, we provide evidence that the superconducting phase of the 2D repulsive Hubbard model derives from the corresponding LE phase with dominant superconducting correlations on ladders with $M$ even legs. In particular, as resumed in the main panel of Fig. 1, we show that the density range where both spin gap and spin parity are finite on ladders extrapolates to the range in which superconductivity is observed in the two-dimensional model.

## 2 Spin parity

In 1D, nonlocal order parameters play a fundamental role in identifying gapped phases of interacting fermions, when spontaneous symmetry breaking by a local order operator, like magnetization, is absent. In the presence of spin-charge separation, it has been shown that each spin/charge gapped phase of correlated electrons has a hidden long-range order, i.e., a finite value of a spin/charge parity or string operator [3,4]. In particular, a finite spin gap signals the emergence of hidden order in the spin channel, which in the LE liquid phase is detected by the finiteness of the expectation value of a spin parity operator $O_s^{1D} = \lim_{L_x \to \infty} \prod_{R=1}^{L_x/2} \exp(2i\pi S_R^z)$, where $S_R^z = \frac{1}{2}(n_{R,\uparrow} - n_{R,\downarrow})$ is the $z$-component of the spin on site $R$. This describes a liquid of holons and doublons in which single electrons appear as bound states (with opposite spins).

Since such state has no reason to disappear when moving to ladders, it is desirable to generalize the definition of spin parity to a higher dimensional geometry. The natural extension to a ladder would be a brane of parities, that includes a further summation over the sites of the rung. Similarly to what has been proposed for describing the Mott phase on ladders in the charge channel [17, 18], we can define:

$$O_s(M, L_x) = \prod_{x=1}^{L_x/2} \exp\left[2i\pi\Delta S(M)\right],\tag{2}$$

where $\Delta S(M) = \sum_{y=1}^{M} S_{x,y}^z$ is the spin fluctuation on the brane with fixed $x$ (here, it is necessary to indicate explicitly the $x$ and $y$ components of the site $R$ in the label of the spin operator, i.e., $S_R^z \equiv S_{x,y}^z$). As in the 1D case, the asymptotic limit with $L_x \to \infty$ must be considered. The quantity defined in Eq. (2) equals 1 in case no single electrons are present and is also unaffected by those electrons which form pairs with opposite spin within the brane; instead, it changes sign whenever one of such pairs crosses the boundary. The longer the length of the boundary (in our case $2M$) the larger is the number of such defects which destroy the presence of order. In particular, in the 2D limit $M \to \infty$ (in addition to $L_x \to \infty$), the expectation value of $O_s(M, L_x)$ would become vanishing, despite the persisting order. Indeed, within a Gaussian approximation one has:

$$\lim_{L_x \to \infty} \langle O_s(M, L_x) \rangle \approx \exp\left\{-2\pi^2 \langle [\Delta S(M)]^2 \rangle\right\},\tag{3}$$

where $\langle \dots \rangle$ indicates the expectation value over the ground-state wave function. In this case, we have that $\lim_{M \to \infty} \langle [\Delta S(M)]^2 \rangle = \infty$. The problem can be solved upon properly renormalizing the prefactor at the exponent of Eq. (2) with the number of legs, and introducing the fractional spin parity order as:

$$C_s(M, L_x) = \langle O_s(M, L_x)^{\frac{1}{M}} \rangle.\tag{4}$$

We mention that, with respect to Ref. [18], for simplicity here we only consider the case in which the exponent of $M$ is equal to 1. For convenience, we also denote as

$$C_s(M) = \lim_{L_x \to \infty} C_s(M, L_x)\tag{5}$$

the $L_x \to \infty$ limit of the fractional spin parity. With the definitions of Eqs. (4) and (5), we obtain that the fractional spin parity remains finite also in its 2D extrapolation $C_s(\infty)$ within the spin gapped phase. Notice that the vanishing of $C_s(\infty)$ in the disordered, i.e., spin gapless, phase is not affected by the fractional definition of the spin parity [18, 35]. To this end, we emphasize that, when extrapolating within the gapless phase, the limit for $L_x \to \infty$ has to be performed before the limit $M \to \infty$.

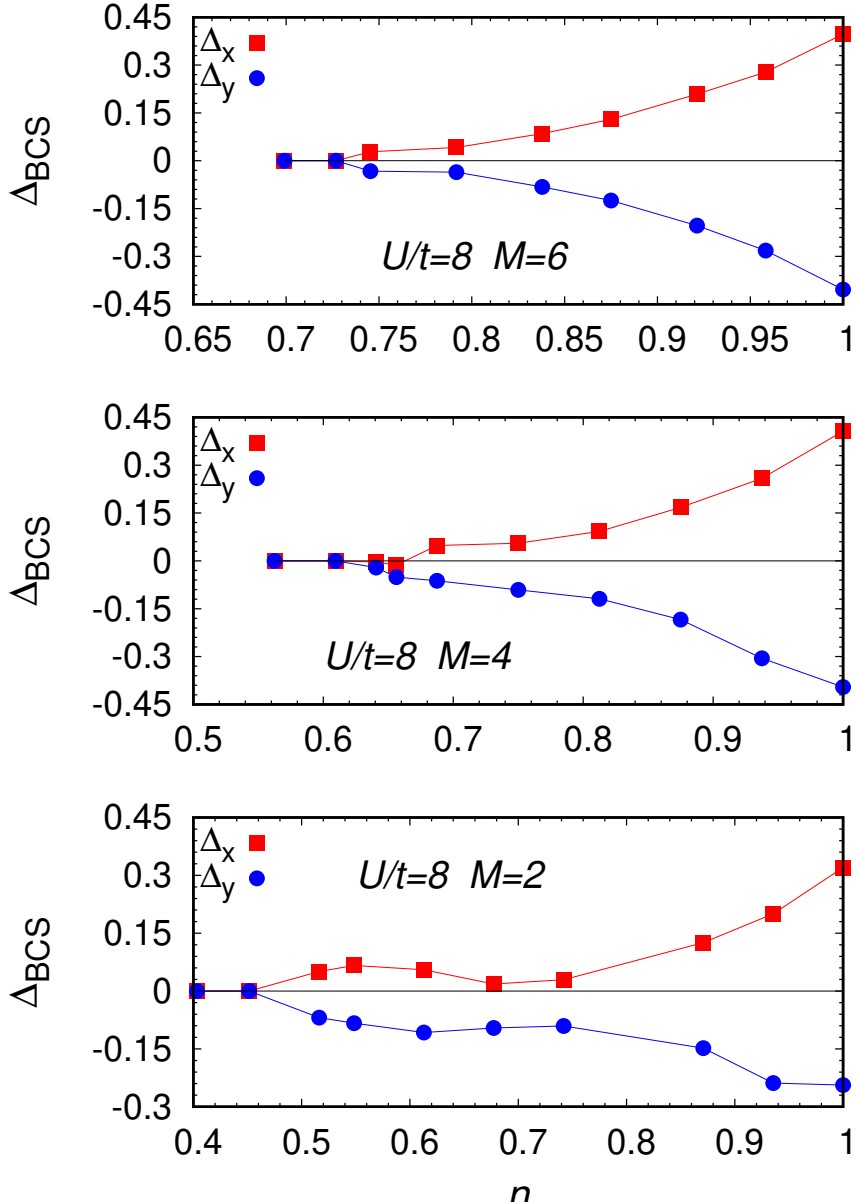

Figure 2: BCS pairing terms $\Delta_x$ (red squares) and $\Delta_y$ (blue circles) as a function of the electron density $n$. Data are reported at $U/t = 8$ for a two-leg system of size $L = 124$ (lower panel), a four-leg system of size $L = 256$ (middle panel), and a six-leg system of size $L = 432$ (upper panel). Error bars are smaller than the symbol size.

In summary, the fractional parity remains finite also in the 2D limit whenever the pairs of electrons with opposite spin have finite correlation length. As anticipated in the inset of Fig. 1, we will see that this latter feature is connected to the presence of $d$-wave superconductivity.

## 3   Variational Monte Carlo method

Our numerical results are obtained by means of the variational Monte Carlo method, which is based on the definition of suitable wave functions to approximate the ground-state properties

beyond perturbative approaches [36]. In particular, we consider the so-called Jastrow-Slater wave functions that extend the original formulation proposed by Gutzwiller to include correlations effects on top of uncorrelated states [37, 38]. Our variational wave functions are described by:

$$|\Psi\rangle = \mathcal{J}|\Phi_0\rangle, \tag{6}$$

where $|\Phi_0\rangle$ is an uncorrelated state that corresponds to the ground state of the following uncorrelated Hamiltonian [39, 40]:

$$\mathcal{H}_{\text{MF}} = \sum_{k,\sigma} \xi_k c_{k\sigma}^\dagger c_{k\sigma} + \sum_k \Delta_k c_{k\uparrow}^\dagger c_{-k\downarrow}^\dagger + \text{h.c.}, \tag{7}$$

which includes a free-band dispersion relation $\xi_k = \epsilon_k - \mu$, where $\epsilon_k$ is the band structure of Eq. (1) with $U = 0$ and $\mu$ is the chemical potential, as well as a BCS coupling $\Delta_k$, with pairings $\Delta_x$ and $\Delta_y$ along the $x$ and the $y$ direction, respectively. The explicit form of the band dispersion is $\epsilon_k = -2t(\cos k_x + \cos k_y)$, for the four-leg and the six-leg ladders. In the case with two legs, the bond along the $y$ direction must be counted only once, thus leading to a slightly different form, namely, $\epsilon_k = -2t \cos k_x \pm t$. Similarly, $\Delta_k = 2(\Delta_x \cos k_x + \Delta_y \cos k_y)$ for the four-leg and the six-leg ladders and $\Delta_k = 2\Delta_x \cos k_x \pm \Delta_y$ for the case with two legs. The parameters $\Delta_x$, $\Delta_y$, and $\mu$ are optimized to minimize the variational energy (while $t = 1$ sets the energy scale of the uncorrelated Hamiltonian). The presence of the BCS pairings allows us to generate a finite spin gap in the variational state on ladder systems, as shown in the next section, while it leads to superconductivity in the 2D case [13, 16, 41]. The effects of correlations are introduced by means of the so-called Jastrow factor $\mathcal{J}$ [42, 43]:

$$\mathcal{J} = \exp\left(-\frac{1}{2}\sum_{R,R'} v_{R,R'} n_R n_{R'}\right), \tag{8}$$

where $n_R = \sum_\sigma n_{R,\sigma}$ is the electron density on site $R$ and $v_{R,R'}$ (that include also the local Gutzwiller term for $\mathbf{R} = \mathbf{R}'$) are pseudopotentials that are optimized for every independent distance $|\mathbf{R} - \mathbf{R}'|$. In particular, the Jastrow factors are crucial to describe the Mott insulator at half filling.

While the model is insulating only at half filling and metallic at any finite doping, a spin gap is present in a large region of doping close to half filling. In order to assess the gapped or gapless nature of the spin excitations, we calculate the spin-spin structure factor $S(\mathbf{q})$, defined as:

$$S(\mathbf{q}) = \frac{1}{L}\sum_{R,R'} \langle S_R^z S_{R'}^z\rangle e^{i\mathbf{q}\cdot(\mathbf{R}-\mathbf{R}')}, \tag{9}$$

where $\langle\dots\rangle$ indicates the expectation value over the variational wave function. In analogy with the relation between the density structure factor and the charge gap [44, 45], we have that a spin gap $\Delta_s$ is present whenever $S(\mathbf{q}) \propto |\mathbf{q}|^2$ for $|\mathbf{q}| \to 0$, while gapless spin excitations are present for $S(\mathbf{q}) \propto |\mathbf{q}|$, since $\Delta_s \propto \lim_{\mathbf{q}\to 0} |\mathbf{q}|^2/S(\mathbf{q})$. In the following, we consider the quantity:

$$E_s = \lim_{\mathbf{q}\to 0} \frac{|\mathbf{q}|^2}{S(\mathbf{q})}, \tag{10}$$

to detect the presence/absence of the spin gap.

Finally, pair-pair correlations can be computed by calculating a correlation function between singlets on rungs at distance $x$, defined as

$$D(x) = \langle \Delta(x+1)\Delta^\dagger(1)\rangle, \tag{11}$$

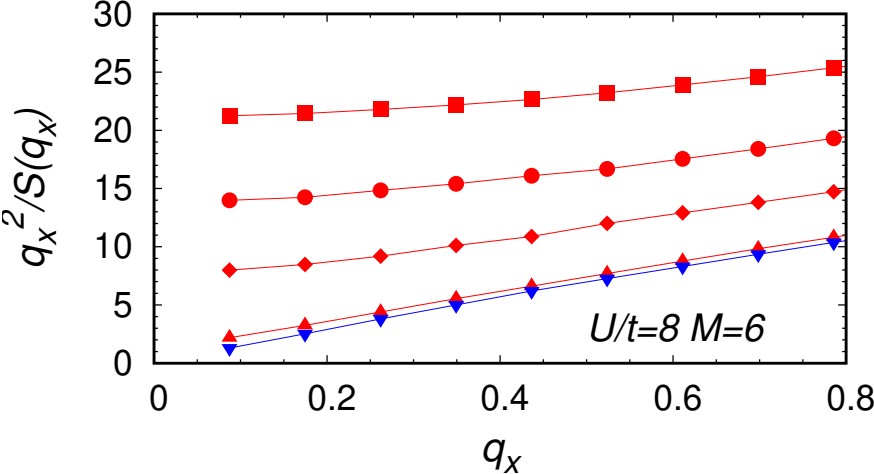

Figure 3: $q_x^2/S(q_x)$ [which is proportional to the spin gap, see Eq. (10)] as a function of $q_x$ at $q_y = 0$. Data are reported at $U/t = 8$ for a six-leg system of size $L = 432$. Red points represent dopings where the spin gap is finite: $n = 1$ (squares), $n = 0.958$ (circles), $n = 0.894$ (diamonds), $n = 0.773$ (up triangles), while blue points represent a case where there is no spin gap: $n = 0.727$ (down triangles). Error bars are smaller than the symbol size.

where

$$\Delta^{\dagger}(x) = c_{x,1,\uparrow}^{\dagger} c_{x,2,\downarrow}^{\dagger} - c_{x,1,\downarrow}^{\dagger} c_{x,2,\uparrow}^{\dagger} \tag{12}$$

is a vertical singlet located on the rung between sites of coordinates $(x, 1)$ and $(x, 2)$. As before, here we explicitly denoted the coordinates of the site $R$, i.e., $c_{R,\sigma}^{\dagger} \equiv c_{x,y,\sigma}^{\dagger}$.

Our simulations are performed with periodic boundary conditions along the $x$ direction, while, in the $y$ direction, they are taken to be open for $M = 2$, antiperiodic for $M = 4$, and periodic for $M = 6$. For $M = 2$, open boundary conditions along the rungs are necessary, in order to avoid a double counting of the intra-rung bonds. For $M = 4$ and $M = 6$, the choice of boundary conditions is dictated by the condition of having a unique and well defined uncorrelated state $|\Phi_0\rangle$ at half filling. Here, the particle-hole symmetry imposes $\mu = 0$ in the free-band dispersion of Eq. (7); since the optimal state has BCS pairing with $d$-wave symmetry, there are four points in reciprocal space, i.e., $k = (\pm\pi/2, \pm\pi/2)$, where the Hamiltonian of Eq. (7) has zero eigenvalues, thus leading to a degenerate ground state. Our choice of boundary conditions is done in order to avoid having these points in reciprocal space.

# 4 Results

First of all, we report in Fig. 2 the optimal value of the variational BCS parameters $\Delta_x$ and $\Delta_y$, as a function of the electron density $n$. The symmetry of the two parameters resembles the $d$-wave one, being $\Delta_y \simeq -\Delta_x$, as expected for a ladder system that becomes a square lattice when the number of legs equals the number of rungs.

Then, we investigate the presence of a spin gap, by looking at the behavior at small momenta of the spin-spin correlations defined in Eq. (9). As an example, we plot in Fig. 3 the quantity $q_x^2/S(q_x)$, as a function of $q_x$ at $q_y = 0$, for the six-leg case. If this quantity extrapolates to a finite number, the system has a spin gap, otherwise it is gapless. Then, in Fig. 4, we report the extrapolation of $q_x^2/S(q_x)$ to the $q_x = 0$ limit, denoted as $E_s$, for ladders with two,

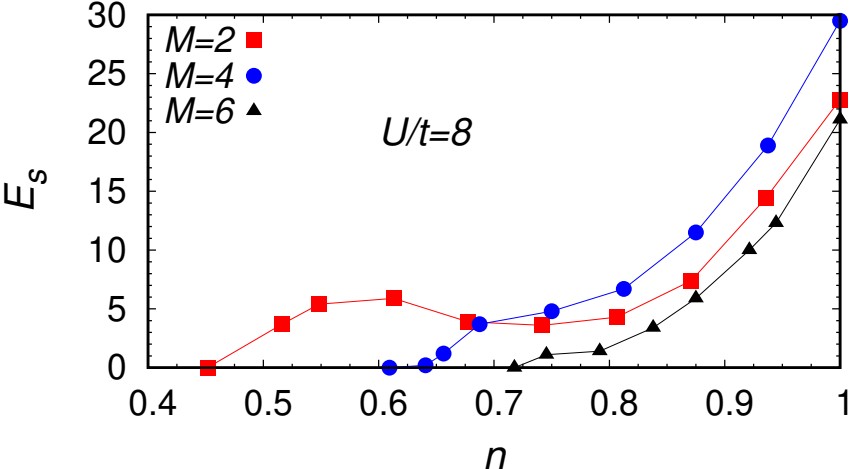

Figure 4: Extrapolation of $q_x^2/S(q_x)$ to $q_x = 0$ (for $q_y = 0$), denoted by $E_s$. Data are reported at $U/t = 8$ for a two-leg system of size $L = 124$ (red squares), a four-leg one of size $L = 256$ (blue circles) and a six-leg one of size $L = 432$ (black triangles). Error bars are smaller than the symbol size.

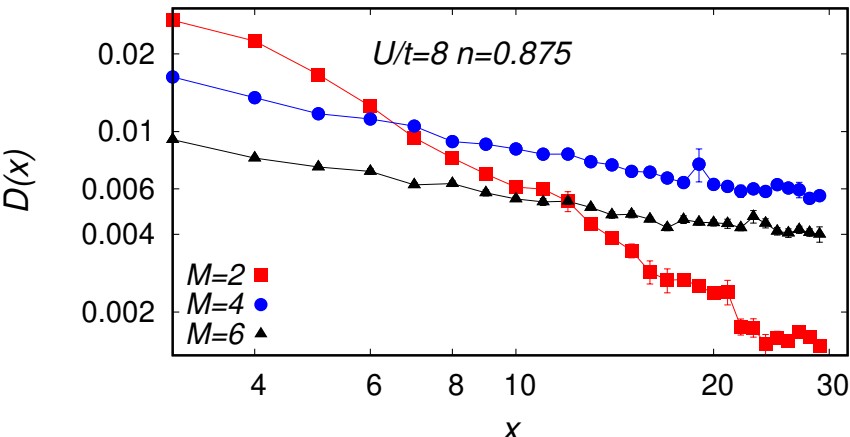

Figure 5: Pair-pair correlations $D(x)$ as a function of the distance $x$ at $n = 0.875$ and $U/t = 8$ for a two-leg ladder (red squares), a four-leg one (blue circles) and a six-leg one (black triangles). Data are shown on a log-log scale in order to highlight the power-law decay.

four, and six legs. The lattice sizes are large enough to be at the thermodynamical limit. We remark that the quantity $E_s$ is only proportional to the spin gap, thus not providing a quantitative estimate of it. In particular, the proportionality constant depends on the geometry of the lattice, thus not allowing for a direct comparison of the gap size between lattices with a different number of legs. Nonetheless, our results show that the gap is finite in the region where the variational parameters $\Delta_x$ and $\Delta_y$ are finite, while it vanishes when the pairing terms are not present in the optimal variational state. Indeed, a finite spin gap is associated to a BCS gap with no gapless points in the wave function. Given the optimal $d$-wave nature of the pairing, this condition is satisfied with the boundary conditions described in the previous paragraph, that do not include the points $(\pm\pi/2, \pm\pi/2)$ in reciprocal space. Finally, note that while the spin gap decreases monotonically to zero for increasing doping in the four-leg and

six-leg systems, it is non monotonic in the two-leg case.

In Fig. 5, we plot the pair-pair correlations $D(x)$ defined in Eq. (11) for two-, four-, and six-leg ladders at $n = 0.875$. Data are shown on a log-log-scale in order to highlight the power-law decay typical of ladder systems: $D(x) \propto x^{-\gamma}$. The exponent $\gamma$ changes from approximately 1.45 for $M = 2$ to approximately 0.34 for $M = 6$. Even if a precise determination of the exponent $\gamma$ is hard, as highlighted by DMRG in the two-leg case [21, 22], nonetheless our results indicate that pair-pair correlations become stronger and stronger as $M$ increases, in agreement with a finite value of $\lim_{x \to \infty} D(x)$, that is expected in the 2D limit [16].

Let us now consider the fractional spin parity of Eq. (4). Our results, shown in Fig. 6, indicate that the fractional spin parity $C_s(M, L_x)$ increases when approaching half filling, following the behavior of the spin gap, with a value that does not depend on $L_x$. On the contrary, a flat behavior in $C_s(M, L_x)$ occurs in the region where no spin gap is present, with this offset getting smaller and smaller when $L_x \to \infty$. The gray areas in Fig. 6 indicate the doping region where the behavior of the fractional spin parity changes and are in agreement with the opening of the spin gap shown in Fig. 4. In Fig. 7, we show, for the two-leg case, that the almost constant value of $C_s(M, L_x)$ in the spin gapless region indeed scales to zero at increasing linear size $L_x$, with the power-law decay illustrated in Ref. [18]. We would like to point out that, even if there are strong finite size effects in the fractional spin parity, one can still clearly distinguish between two different regimes: A flat and size dependent behavior of $C_s(M, L_x)$ in the spin gapless region and a value of the fractional spin parity that is proportional to the spin gap and that does not depend on the lattice size in the spin gapped region. Our results indicate then that the fractional spin parity can be considered a good indicator for the presence of a spin gap in ladder systems. Moreover, we observe that $C_s(M, L_x)$ is approximately equal in the spin gapped regions of the four-leg and of the six-leg systems, supporting the fact that this quantity remains finite in the 2D limit.

We would like to mention that our picture is still valid when the variational wave function includes the stripe order of Ref. [34]. Indeed, we have optimized at doping 1/8 (i.e., $n = 0.875$), a variational state that combines BCS pairing with stripe order, on different lattice sizes [46]. Even if in such a state both the variational BCS parameters and the fractional spin parity are reduced with respect to the uniform case, they are size independent, thus indicating that the fractional spin parity remains finite in the thermodynamical limit.

The previous results are resumed in the phase diagram anticipated in Fig. 1. There the evolution of the critical density $n_c$ where the spin gap opens, or equivalently where the fractional parity $C_s(M)$ is finite, is reported as a function of the number of legs. Our results show that the density at which the spin gap opens becomes closer to half filling as the number of legs increases, approaching the value where superconductivity develops in a fully 2D numerical simulation ($n \approx 0.75$) [16]. Moreover, in the inset of Fig. 1, we present a tentative extrapolation of the fractional spin parity $C_s(M)$ of Eq. (5) to the 2D case. In the shaded region with $n \gtrsim 0.75$, i.e., where superconductivity is present, $C_s(\infty)$ is extrapolated from the results for the two-, four-, and six-leg cases, that are shown in Fig. 6. In the region with $n \lesssim 0.75$, we expect that the fractional spin parity vanishes in the thermodynamical limit, according to the results presented in Figs. 6 and 7.

# 5 Conclusions

By means of a variational quantum Monte Carlo analysis of the repulsive Hubbard model, we have investigated the possibility that at the origin of both the LE spin gapped phase with dominant superconducting correlations, observed on ladders with even number $M$ of legs, and the 2D $d$-wave superconducting region, there is the same mechanism of formation of pairs of

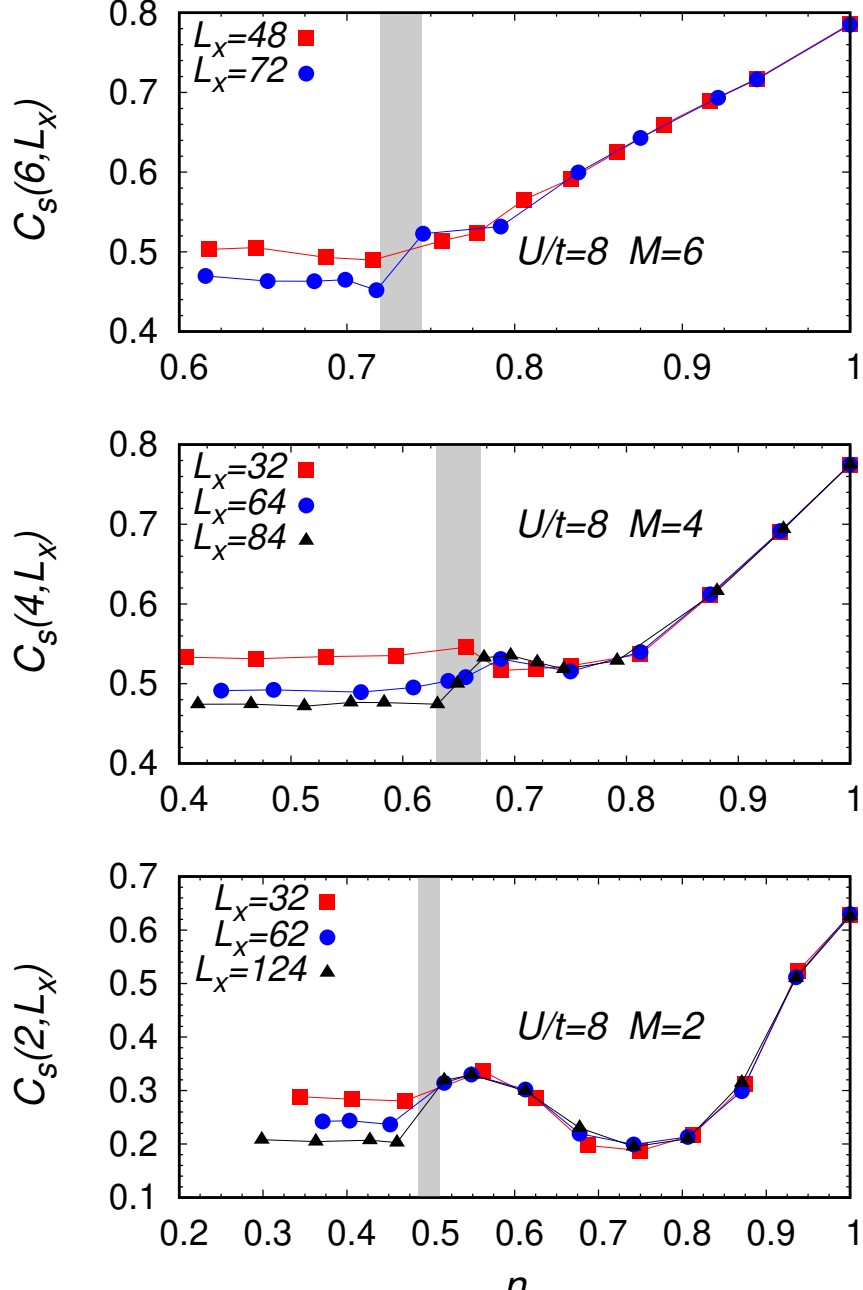

Figure 6: Fractional spin parity $C_s(M, L_x)$, as a function of $n$ at $U/t = 8$. Lower panel: two-leg system for $L = 64$ (red squares), $L = 124$ (blue circles), and $L = 248$ (black triangles) lattice sizes. Middle panel: four-leg system for $L = 128$ (red squares), $L = 256$ (blue circles), and $L = 336$ (black triangles) lattice sizes. Upper panel: six-leg system for $L = 288$ (red squares), and $L = 432$ (blue circles) lattice sizes. The gray boxes denote the dopings where the spin gap opens. Error bars are smaller than the symbol size.

electrons with opposite spin, that have a finite correlation length. This aspect is captured by the hidden ordering of a spin parity operator, which remains finite also in the 2D limit upon an appropriate normalization to $M$.

Our results show a good agreement between the presence of a finite spin gap and a finite

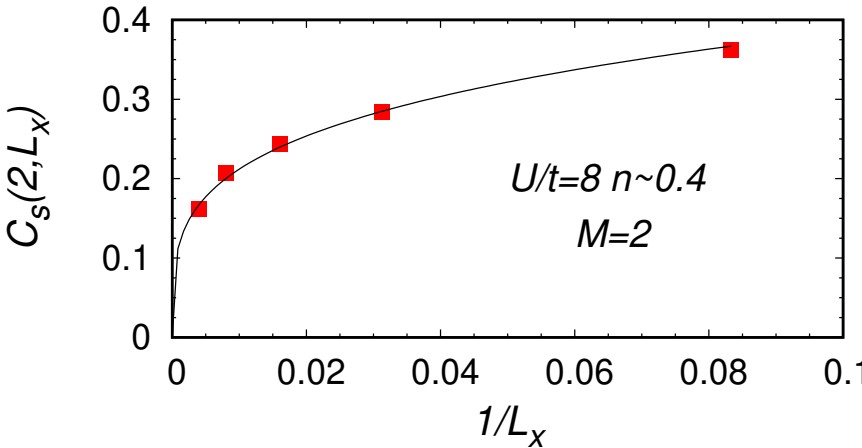

Figure 7: Behavior of the fractional spin parity $C_s(2, L_x)$, as a function of the inverse linear dimension $1/L_x$, for a two-leg ladder, in the spin gapless region ($n \approx 0.4$). Data are shown at $U/t = 8$. The black curve is a fit of the points with the law $C_s(M, L_x) = A(L_x/2)^{-\alpha}$. Error bars are smaller than the symbol size.

value of the fractional spin parity. In particular, the value of the fractional spin parity is size independent and proportional to the size of the gap in the spin gapped phase, while it becomes a size dependent constant in the gapless phase, extrapolating to zero as a power law in the linear dimension of the system. The spin gapped region is characterized by a finite value of the BCS pairing terms in the variational state, with $d$-wave symmetry, and by superconducting pair-pair correlations with a power-law decay. The region where the spin gap (and the fractional spin parity) is finite shrinks with the number of legs, with an extrapolation to the 2D case that matches with the region where superconductivity is detected in the 2D Hubbard model, i.e., where pair-pair correlations remain finite at large distances.

Our observations are thus of interest for understanding the physics of high-$T_c$ superconductivity in cuprate materials, which is believed to be captured by the Hubbard model. Our findings support the idea of a pairing interaction mediated by correlated antiferromagnetic spin fluctuations, as characteristic of the lower dimensional LE phase [47, 48]. It must be stressed that the nature of such phase changes when passing from the attractive to the repulsive case discussed here. In fact, in the attractive case the correlated pairs of electrons with opposite spin on neighboring sites represent minority fluctuations with respect to the majority of holons and doublons. Whereas in the repulsive case they become the majority of electrons. The difference is seen for example in the dependence on the number of legs of the spin fluctuations on the brane, which according to Eqs. (3) and (4) appear to be proportional to $M^2$, at variance with the $M$ dependence typical of the $U < 0$ case [18]. In fact, while in the $U < 0$ case superconducting pairs are on site, for $U > 0$ they are on neighboring sites. At half filling, they coexist with the minority of holon-doublon correlated pairs in the insulating phase. Upon doping, the holon-doublon pairs break while the up-down electron pairs remain correlated. In the 2D limit, this originates a collective behavior characterized by spontaneous symmetry breaking and local order, as the finiteness of pairing correlation proves. This provides an example of how hidden orders may evolve into standard local orders in higher dimension.

# Acknowledgements

We thank M. Fabrizio and C. Gros for useful discussions. We thank C. Degli Esposti Boschi for providing DMRG numerical data.

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
