# Peer review of "Superconductivity in the Hubbard model: a hidden-order diagnostics from the Luther-Emery phase on ladders"

_SciPost Physics, doi:SciPost Phys. 6, 018 (2019)_

## Round 1 · Referee Report · Anonymous · 2018-12-16

Strengths

1) The paper is very clearly written
2) The results provide an interesting insight in the question of emergence of d-vave superconductivity in two dimensions

Weaknesses

1) The paper could contain some more details about the numerics, in particular about some choices made (e.g. boundary conditions, length of the system, …)

Report

The authors discuss the emergence of the Luther-Emery phase in the Hubbard models on ladders with an even number of legs. The problem is tackled with a numerical (variational Monte Carlo) techniques: the ground state of 2-, 4-, 6-legs case is considered, in the repulsive regime and for a wide range of fillings. The analysis is carried out by calculating: i) the spin structure factor which yields information on whether the phase is gapped or gapless ; ii) the pair-pair correlation function between singlets on the rung; iii) the (non-local) spin parity operator. The results show that, similarly to what is expected for the 2 dimensional case, a gapped d-wave like phase is found to be stable at large doping.

The topic considered by the author is very interesting and seems to be very important in order to describe the emergence of superconductivity in the two dimensional case, via a detailed analysis of quasi one-dimensional Hubbard models, defined on spin ladders.
The paper is written in a very clear way and all the results are presented in a concise but precise way.

I believe that the paper deserves publication.

I have only a few minor comments, that the authors might want to consider to better explain their work.

Requested changes

1) Line after eq. (7), the sentence “a free-band dispersion ξk, defined as in the Hubbard Hamiltonian” could be made more explicit;
2) Why simulations are performed with different boundary conditions along the y-direction (see after eq. (12)) depending on the number of legs?
3) Are the sizes L of the systems considered in numerical simulations dictated by the maximum one that computation can achieve or by other considerations? Also, what about a check of the thermodynamical limit?
4) I do not understand the sentence: “no gapless points in the wave function “ at page 9.

  • validity: high
  • significance: high
  • originality: top
  • clarity: top
  • formatting: excellent
  • grammar: excellent

Author:  Luca Fausto Tocchio  on 2018-12-19  [id 391]

(in reply to Report 1 on 2018-12-16)
Category:
answer to question

We thank the Referee for recognizing the quality and the potential impact of our work, as well as the clarity in presenting the results. We provide here a response to his/her questions and comments, in order to better clarify some technical details. The manuscript will be changed accordingly.

1) The free-band dispersion relation $\xi_k=\epsilon_k-\mu$ in the uncorrelated Hamiltonian of Eq.(7) includes the band structure $\epsilon_k$ of the $U=0$ Hamiltonian and the chemical potential $\mu$. The explicit form of the band dispersion is $\epsilon_k=-2t(\cos k_x + \cos k_y)$, for the 4-leg and the 6-leg ladder. In the case with 2 legs, the vertical bond must be counted only once, thus leading to a slightly different form, namely, $\epsilon_k=-2t\cos k_x \pm 1$. We add this information in the revised version of the paper.

2) and 4) These two points raised by the Referee are intimately related. In our approach, the variational wave function is constructed from the ground state of an auxiliary (quadratic) Hamiltonian, i.e., by filling the $N_e$ lowest-energy levels, where $N_e$ is the number of electrons. Then, in order to deal with a unique and well defined state, there must be a gap (that can be just due to finite size effects) between the $N_e$-th and the $(N_e+1)$-th levels. The optimal state contains BCS pairing with $d$-wave-like symmetry; thus the spectrum has gapless points at $k=(\pm \pi/2,\pm \pi/2)$. Our choice of boundary conditions is done in order to avoid having these points in reciprocal space. We clarify this fact in the revised version of the paper.

3) The sizes $L$ that we consider are taken large enough to be close to the thermodynamic limit for the quantities of interest, even if largest lattices can be considered in principle. For instance, in Fig.6, the fractional spin parity in the spin gapped region has a value that does not depend on the size of the lattice, i.e., the results are already in the thermodynamic limit. The fractional spin parity in the spin gapless region is instead still finite for the lattice that we considered; in this respect, we have shown in Fig.7 that the fractional spin parity extrapolates to zero when larger lattices and a proper scaling function are taken into account.

---

## Round 1 · Referee Report · Neil Robinson · 2019-1-3

Strengths

1- Clear presentation of results
2- Well written manuscript
3- Interesting results that draw insights into 2D physics from quasi-1D ladder models

Weaknesses

1- Choice of boundary conditions for the $M=2,4,6$ simulations is unclear
2- Accuracy of the simulations as compared to DMRG/tensor networks could be clearly stated
3- Conclusions about nature of the pairing in 2D could be more clearly stated and discussed in the conclusions

Report

In this work, the authors study $M=2,4,6$-leg Hubbard ladders with a particular focus on the Luther-Emery phase. This phase is characterized by the presence of a spin gap and gapless charge excitations, leading to the possibility of quasi-long-range superconducting order if interactions conspire appropriately. It is well known from numerous studies of ladder models that this superconductivity is d-wave in nature, leading to obvious parallels to the cuprates, which are thought to be described by the two-dimensional Hubbard model. The two-dimensional limit is tough to tackle, but can be pictured as the $M\to\infty$ limit of the studied ladder models. Using variational quantum Monte Carlo (VQMC) the authors study how a "fractional" generalization of the spin parity operator, which captures the formation of correlated electron pairs, varies with the number of legs $M$. They use their results to gain insight into the two-dimensional limit, with the region in which the fractional spin parity operator is finite coinciding with that in which superconductivity is observed in fully two-dimensional calculations.

The manuscript is well written, the presentation of the results is clear, and the conclusions are well justified. I support publication in SciPost Physics provided the following comments are addressed by the authors.

Requested changes

1. In the abstract the authors state "Our observations support the idea that superconductivity emerges out of spin gapped phases on ladders, driven by a spin-pairing mechanism". This observation is of interest with regards to high-temperature superconductivity in the cuprates, and is not so clearly stated in the conclusions. I think it would be good to state (and possibly discuss) this explicitly there, and it may also be worth noting that similar ideas lie at the heart of the Yang-Rice-Zhang ansatz for the single-electron propagator in the cuprates (see, e.g., the review Rep. Prog. Phys. 75 016502 (2012)) and recent works by Tsvelik on the cuprate problem (see, e.g., Phys. Rev. B 95, 201112 (2017)).

2. As mentioned by the authors in the introduction, the density matrix renormalization group (DMRG) has been used extensively to study $M$-leg ladders. How does the presented VQMC compare to DMRG for, e.g., the value of the ground state energy or for reproducing the phase diagram as a function of filling and U?

3. As shown explicitly in Ref. [28] by Lin, Balents and Fisher, many details of the $M>2$ phase diagram depend on the boundary conditions along the rungs. Following Eq. (12), the authors state the boundary conditions they use for $M=2$ (open), $M=4$ (antiperiodic) and $M=6$ (periodic), but its unclear why these choices were made. In light of Figs. 9 and 10 of Ref. [28] for $M=4$, this needs to be discussed and justified. Also, for $M=2$ aren't open and periodic boundary conditions equivalent?

4. It would also be good to mention somewhere about the statistical errors. If the error bars are smaller than the points, this should be stated explicitly.

5. More details on the extrapolation to the $M=\infty$ limit would be welcome - is the "2D limit" shown in Fig. 1 from an extrapolation or fully 2D numerical calculations?

  • validity: high
  • significance: high
  • originality: high
  • clarity: top
  • formatting: perfect
  • grammar: excellent

Author:  Luca Fausto Tocchio  on 2019-01-23  [id 409]

(in reply to Report 2 by Neil Robinson on 2019-01-03)
Category:
answer to question

We thank the Referee for recognizing the relevance of our work in drawing insight into the 2D physics, as well as the clarity in presenting the results. We provide here a response to his questions and comments. The manuscript will be changed accordingly.

1) We thank the referee for pointing out that, unlike in the abstract and along the text, the possible relevance of our results to high-Tc superconductivity in cuprates was not addressed in the conclusion. We have now added a sentence at the beginning of the last paragraph in which we discuss this aspect and we have made reference to the recent literature cited by the referee on the topic.

2) A detailed comparison between the variational Monte Carlo and DMRG techniques has been reported in LeBlanc et al., PRX 5, 041041 (2015) for the 2D case. In the submitted manuscript, we use a simplified variational wave function without backflow correlations, which gives a slightly less accurate estimation of the ground-state energy. Still, correlation functions are only weakly affected by the presence/absence of backflow correlations, see Tocchio et al., PRB 83, 195138 (2011), and we decided, for this reason, not to include the computationally demanding backflow correlations in our variational state.

A comparison between VMC and DMRG results in the one-dimensional case (i.e., $M=1$) has been performed in Tocchio et al., PRB 81, 205109 (2010), showing a very good agreement.

For ladders with $M>1$, a detailed comparison between VMC and DMRG does not exist. On finite clusters, DMRG performs at best with open boundary conditions (OBC) along the legs, while VMC prefers periodic boundary conditions (PBC) because wave functions are translationally invariant. Therefore, as in the 2D case, the comparison should be done in the thermodynamic limit. At the moment, accurate extrapolations do not exist for generic values of $U/t$, electronic densities, and number of legs $M$. By contrast, calculations with PBC within DMRG are possible for small systems. Then, we can perform a comparison between our results for fractional spin parity, as well as for fractional charge parity, and data reported in Ref.[17] at half filling for a 2-leg ladder with $L=18$ sites. The attached figure shows that we obtain a very good agreement for fractional charge parity, while we slightly overestimate the DMRG results for fractional spin parity, even if the qualitative trend is the same in both approaches.

Finally, we would like to mention that accurate DMRG results are not easy to get in the generic case with $M>2$, since they are computationally demanding, and that even for $M=2$ some discrepancies have been obtained in the past, see the difference in the exponent of the power-law decay of superconducting correlations in Ref.[21] and in Ref.[22].

3) For $M=2$, open boundary conditions along the rungs are necessary, in order to avoid a double counting of the intra-rung bonds.

For $M=4$ and $M=6$, the choice of boundary conditions is dictated by the condition of having a unique and well defined uncorrelated state $|\Phi_0\rangle$ at half filling. Here, the particle-hole symmetry imposes $\mu=0$ in the free-band dispersion of Eq.(7); since the optimal state has BCS pairing with $d$-wave symmetry, there are four points in reciprocal space, i.e., $k=(\pm \pi/2,\pm \pi/2)$, where the Hamiltonian of Eq.(7) has zero eigenvalues, thus leading to a degenerate ground state. Our choice of boundary conditions is done in order to avoid having these points in reciprocal space.

We recognize that this point was not clear enough in the previous version of the manuscript and we will clarify it better in the new version.

4) We thank the Referee for having raised this issue. The error bars are smaller than the symbol size, except in Fig. 5 (where superconducting correlations are shown). We update the manuscript inserting the error bars in Fig. 5 and stating explicitly that error bars are smaller than the symbol size, when not shown.

5) The 2D-limit presented in Fig.1 is a full 2D calculation. This result is taken from Ref.[16] and corresponds to the density at which pairing correlations become negligible in two dimensions, within a Variational Monte Carlo approach. This information is now included in the new version of the manuscript.

Attachment:

Parity_comparison.pdf

---

## Round 2 · Author Response

Dear Editor,

we resubmit the revised version of the manuscript, where we addressed the questions and comments raised by the referees.
We already provided a reply to both referees and we will give a point-by point list of changes in the next section.

Your sincerely,
Luca F. Tocchio, on behalf of all the authors

---

## Round 2 · List of Changes

-) The affiliation of the second author is now changed.
-) In Fig.1, we clarified that the value of the critical density where superconducting correlations start to develop in the 2D case
is obtained from a fully 2D numerical calculation.
-) We included the error bars in Fig. 5 and we specified that the error bars are smaller than the symbol size in Figs. 2, 3, 4, 6, and 7.
-) We wrote explicitly the band dispersion and the pairing terms in Eq.(7).
-) We clarified the choice of boundary conditions, at the end of the Variational Monte Carlo method section.
-) We expanded the last paragraph of the conclusions and we included two extra references (Refs. 47 and 48),
following the suggestion of the second referee.
-) We updated the acknowledgements.

---

## Editorial Decision

published